# Thymosin *β*4 promotes autophagy and repair via HIF-1*α* stabilization in chronic granulomatous disease

Giorgia Renga[1],*, Vasilis Oikonomou[1],*, Silvia Moretti[1], Claudia Stincardini[1], Marina M Bellet[1], Marilena Pariano[1], Andrea Bartoli[1], Stefano Brancorsini[1], Paolo Mosci[2], Andrea Finocchi[3], Paolo Rossi[3], Claudio Costantini[1]📷, Enrico Garaci[4], Allan L Goldstein[5], Luigina Romani[1]📷

Chronic granulomatous disease (CGD) is a genetic disorder of the NADPH oxidase characterized by increased susceptibility to infections and hyperinflammation associated with defective autophagy and increased inflammasome activation. Herein, we demonstrate that thymosin *β*4 (T*β*4), a g-actin sequestering peptide with multiple and diverse intracellular and extracellular activities affecting inflammation, wound healing, fibrosis, and tissue re-generation, promoted in human and murine cells noncanonical autophagy, a form of autophagy associated with phagocytosis and limited inflammation via the death-associated protein kinase 1. We further show that the hypoxia inducible factor-1 (HIF-1)*α* was underexpressed in CGD but normalized by T*β*4 to promote autophagy and up-regulate genes involved in mucosal barrier protection. Accordingly, inflammation and granuloma formation were impaired and survival increased in CGD mice with colitis or aspergillosis upon T*β*4 treatment or HIF-1*α* stabilization. Thus, the promotion of endogenous pathways of inflammation resolution through HIF-1*α* stabilization is drug-gable in CGD by T*β*4.

## Introduction

Chronic granulomatous disease (CGD) is an immunodeficiency caused by mutations in the proteins forming the NADPH complex, which results in defective production of reactive oxygen species (ROS), impaired microbial killing by phagocytic cells, and increased susceptibility to infections (Rider et al, 2018). A common feature of CGD patients is the presence of a hyperinflammatory state in multiple organs, including the gastrointestinal and urogenital tract, lungs, and eyes (Rider et al, 2018), to which inflammation caused by defective LC3-associated phagocytosis (LAP) greatly contributes (de Luca et al, 2014).

LC3-associated phagocytosis is a noncanonical autophagy pathway that plays a key role in the process of linking signals from phagocytosis to inflammation and innate immune responses (Henault et al, 2012; Martinez et al, 2016). Different from canonical autophagy, LAP is activated during phagocytosis upon recognition of microbes by pattern recognition receptors for rapid pathogen degradation (Simon & Clarke, 2016; Sprenkeler et al, 2016). The efficient clearance of the infectious products promoted by LAP could by itself be sufficient to reduce the inflammatory response and, hence, immunopathology. However, a mechanism by which inflammation is regulated during LAP has been recently described and involves the death-associated protein kinase 1 (DAPK1) (Oikonomou et al, 2016), a kinase mediating many different cellular functions such as cell death and repair (Bialik & Kimchi, 2006; Singh et al, 2016). Activated by IFN-γ, DAPK1 not only mediates LAP of the fungus *Aspergillus fumigatus* but also concomitantly inhibits nod-like receptor protein 3 (NLRP3) activation, thus restraining pathogenic inflammation (Oikonomou et al, 2016). Of interest, DAPK1 activity was defective in murine and human CGD (Oikonomou et al, 2016), a finding suggesting that the LAP/DAPK1 axis may represent a druggable pathway in CGD (Oikonomou et al, 2018).

Besides participating in direct microbial killing, the generation of ROS by the influx of neutrophils during infection is accompanied by local oxygen consumption that results in a condition known as inflammatory hypoxia, with stabilization of the hypoxia inducible factor-1 (HIF-1)*α* and resolution of inflammation (Campbell et al, 2014). This phenomenon is particularly relevant in the colonic mucosa, and the effect of HIF-1*α* in the induction of angiogenesis- and glycolysis-related genes as well as genes involved in mucosal barrier protection has been validated in animal models of colitis and in human-derived colonic tissue (Campbell & Colgan, 2019). Consistent with the role of ROS in inflammatory hypoxia, most CGD

[1]Department of Experimental Medicine, University of Perugia, Perugia, Italy   [2]Internal Medicine, Department of Veterinary Medicine, University of Perugia, Perugia, Italy   [3]Department of Pediatrics, Unit of Immune and Infectious Diseases, Children's Hospital Bambino Gesù, Rome, Italy   [4]University San Raffaele and Istituto di Ricovero e Cura a Carattere Scientifico San Raffaele, Rome, Italy   [5]Department of Biochemistry and Molecular Medicine, the George Washington University, School of Medicine and Health Sciences, Washington, DC, USA

Correspondence: luigina.romani@unipg.it
*Giorgia Renga and Vasilis Oikonomou contributed equally to this work

patients manifest inflammatory bowel disease–like symptoms (Campbell & Colgan, 2019), and pharmacological stabilization of HIF-1α within the mucosa protected CGD mice from severe colitis (Campbell et al, 2014). Although the contribution of inflammatory hypoxia in the lung is disputed (Taylor & Colgan, 2017), hypoxia develops during pulmonary invasive fungal infection in models of invasive aspergillosis, including CGD mice (Grahl et al, 2011), and HIF-1α stabilization is required for protection (Shepardson et al, 2014). Of note, HIF-1α mediates the autophagic process induced by a hypoxic environment (Bellot et al, 2009), thus raising the interesting hypothesis that a defective HIF-1α induction/stabilization in CGD patients might be causally related to the impaired autophagy and that pharmacological stabilization of HIF-1α might restore LAP/DAPK1 and immune homeostasis during infection with CGD.

Thymosin β4 (Tβ4) is a major g-actin sequestering peptide found in eukaryotic cells and represents 70–80% of the total thymosin content in human tissues. It is an active peptide with 43 amino acids with moonlighting properties and multiple and diverse intracellular and extracellular activities (Goldstein et al, 2005). Several physiological properties of Tβ4 have been reported, including the regulation of wound healing, inflammation, fibrosis, and tissue regeneration (Goldstein et al, 2012). The circumstantial evidence points to Tβ4 as a potential molecule that could link HIF-1α stabilization to LAP in CGD. First, Tβ4 promotes HIF-1α stabilization (Oh et al, 2008; Jo et al, 2010; Oh & Moon, 2010) and, in turn, HIF-1α may transcriptionally regulate Tβ4 expression (Ryu et al, 2014). Second, Tβ4 is the major actin-sequestering molecule in all eukaryotic cells (Ballweber et al, 2002) and the actin networks contribute to autophagosome formation and membrane remodeling during autophagy (Aguilera et al, 2012), which suggests a possible role for Tβ4 in autophagy.

Based on these premises, in the present study, we have resorted to in vitro and in vivo studies involving human cells and mice with CGD to provide evidence that Tβ4 promotes LAP involving DAPK1 in murine and human CGD although concomitantly impairing granuloma formation in the lung and gut of mice with CGD. Both autophagy and repair were dependent on HIF-1α stabilization, in the lung and likely in the gut, a finding qualifying Tβ4 as a promising peptide with beneficial effects in CGD through the promotion of endogenous pathways of autophagy and inflammation resolution.

# Results

### Tβ4 promotes LC3-associated phagocytosis involving DAPK1

In order to assess the ability of Tβ4 to promote autophagy, we first evaluated the ratio of LC3-II to LC3-I, widely used to monitor autophagy (Oikonomou et al, 2016), on RAW264.7 cells exposed to live Aspergillus conidia in the presence of different concentrations of Tβ4. We have already shown that internalized conidia undergo swelling and concomitantly induce autophagy in these cells (de Luca et al, 2014; Oikonomou et al, 2016). Immunoblotting revealed that Tβ4, although not inducing autophagy in unpulsed cells (Fig 1A), dose-dependently increased the LC3-II to LC3-I ratio in cells pulsed with conidia, an effect observed as early as 2 h after the exposure to the fungus (Fig 1B). This finding suggests that Tβ4 could

be able to activate LAP. To confirm this, the expression of DAPK1 and Rubicon proteins, known to be involved in LAP (Martinez et al, 2016), was dose-dependently increased by Tβ4 (Fig 1C), a finding suggesting that Tβ4 promotes noncanonical autophagy involving DAPK1 and Rubicon. This result prompted us to assess the ability of Tβ4 to restore LAP in CGD, in vitro and in vivo. To this purpose, we purified macrophages from the lungs of C57BL/6 and $p47^{phox-/-}$ mice and pulsed them in vitro with A. fumigatus conidia in the presence of Tβ4. Consistent with previous findings (de Luca et al, 2014; Oikonomou et al, 2016), both autophagy and DAPK1 expression were defective in the cells from $p47^{phox-/-}$ mice but were dose-dependently restored by Tβ4 (Fig 1D). Confirming the murine observation, Tβ4 also increased LC3B expression in monocytes from CGD patients exposed to Aspergillus conidia in vitro (Fig 1E), a finding suggesting that Tβ4 is able to restore LAP involving DAPK1 in human CGD. For in vivo, we resorted to two different experimental models that mimic the human pathology, such as lung and gut inflammation. To this purpose, we infected C57BL/6 and $p47^{phox-/-}$ mice with A. fumigatus intranasally and treated them with Tβ4 for 7 consecutive days starting a week after the infection. LC3-II (Fig 1F) and DAPK1 (Fig 1G) expression were both defective in $p47^{phox-/-}$ mice but restored by Tβ4 and ablated (LC3-II) upon siTβ4 (Fig S1). For gut inflammation, we resorted to the acute colitis in $p47^{phox-/-}$ mice by administering 2.5% dextran sulfate sodium (DSS) in drinking water for 7 d followed by 7 d of DSS-free autoclaved water. Tβ4 was therapeutically administered daily for 7 d, after DSS treatment, at the time at which mice started to lose weight. The DSS treatment has been reported to repress Dapk1 gene expression in colonic epithelial cells (Takeshima et al, 2012). Consistent with previous findings (de Luca et al, 2014), LC3-II (Fig 1H) and DAPK1 (Fig 1I) expression were both defective in the colon of $p47^{phox-/-}$ mice with colitis as opposed to WT mice but restored upon treatment with Tβ4 (Fig 1H and I).

### Tβ4 promotes HIF-1α expression in CGD

Given that the defective LAP in CGD is amenable to restoration by Tβ4, we wonder whether the production of Tβ4 could be defective in CGD. To this purpose, we assessed Tβ4 gene and protein expression in $p47^{phox-/-}$ mice. We found a lower expression of Tβ4 in CGD lungs than that in C57BL/6 mice, both in terms of gene and protein expression, as revealed by real-time (RT) PCR and immunoblotting (Fig 2A) and confirmed by immunofluorescence staining (Fig 2B). Given the reciprocal regulation between Tβ4 and HIF-1α (Oh et al, 2008; Ryu et al, 2014), we assessed whether defective Tβ4 levels in CGD mice could be associated with altered HIF-1α expression. This turned out to be the case, as HIF-1α levels were reduced in CGD mice (Fig 2C and D) and Tβ4 ablation decreased HIF-1α in C57BL/6 mice (Fig S1A). Interestingly, administration of Tβ4 could restore HIF-1α levels in CGD mice (Fig 2E and F), whereas HIF-1α silencing decreased Tβ4 expression (Fig 2B). Defective Tβ4 expression (Fig 2G) and restoration of HIF-1α expression upon administration of Tβ4 (Fig 2H) were also observed in the colon. Consistent with the murine results, Tβ4 also increased HIF-1α expression in monocytes from CGD patients challenged with Aspergillus conidia (Fig 2I), thus suggesting that Tβ4 is able to restore HIF-1α expression in human CGD. These results indicate that intracellular autocrine crosstalk between Tβ4 expression and HIF-1α induction occurs in CGD.

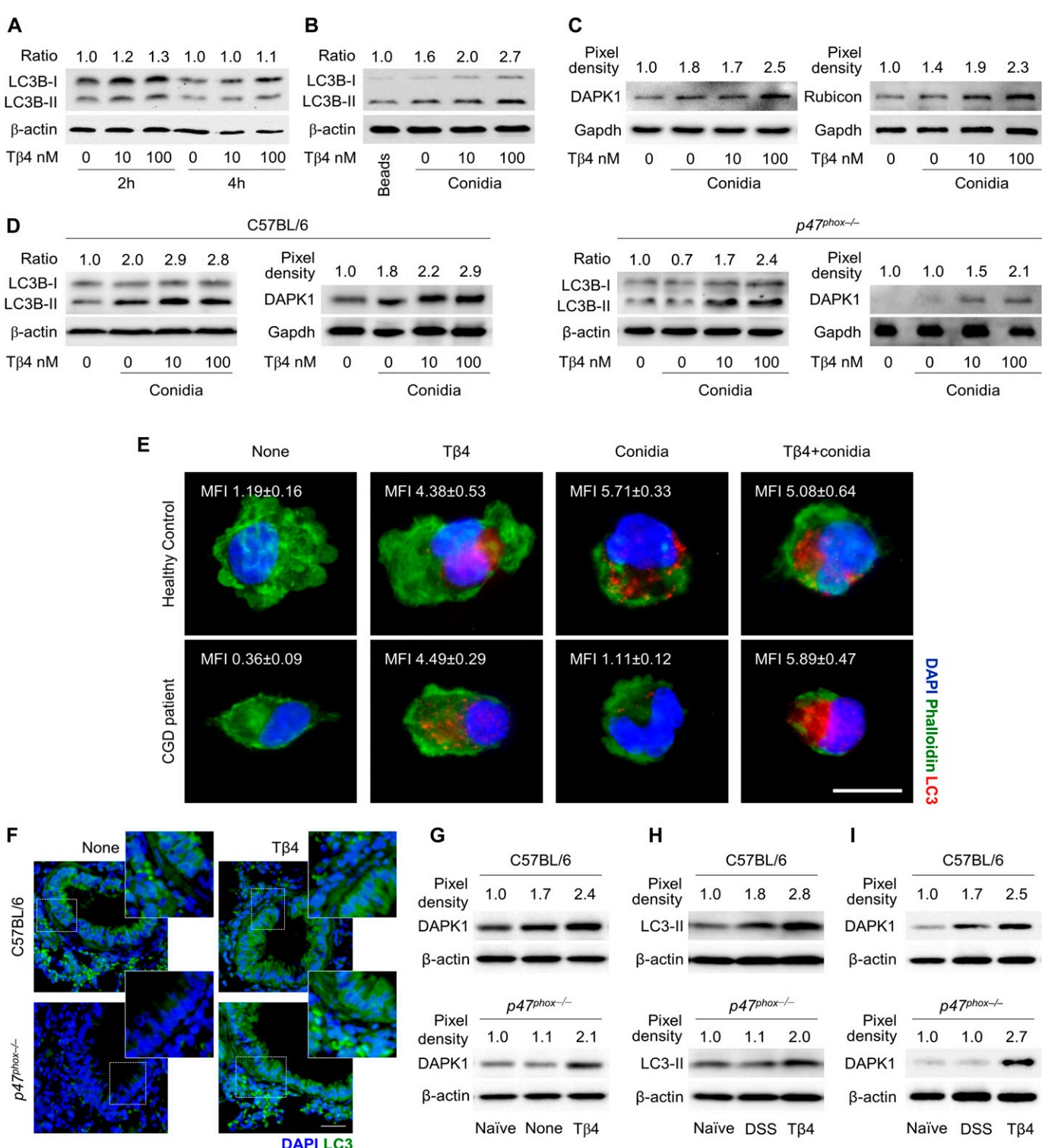

**Figure 1. Tβ4 promotes LC3-associated phagocytosis involving DAPK1.**

**(A)** LC3B-II/LC3B-I expression in RAW264.7 cells after 2 or 4 h stimulation with 10 and 100 nM Tβ4. **(B, C)** LC3B-II/LC3B-I and (C) DAPK1 and Rubicon expression in RAW264.7 cells pulsed for 2 h with *A. fumigatus* conidia after 1-h pretreatment with 10 and 100 nM of Tβ4. In (B), inert beads were used as the control. **(D)** LC3B-II/LC3B-I and DAPK1 production in lung macrophages from C57BL/6 and *p47phox−/−* mice after pulsing with *A. fumigatus* conidia in the presence of Tβ4. **(E)** LC3 expression on monocytes from CGD patients or healthy controls pretreated with 100 nM Tβ4 and stimulated for 2 h with the fungus. **(F, G)** LC3 and (G) DAPK1 expression on the lung of C57BL/6 and *p47phox−/−* mice infected intranasally with *A. fumigatus* conidia and treated i.p. with 5 mg/kg Tβ4 for 7 consecutive days starting a week after the infection. **(H, I)** LC3 and (I) DAPK1 expression in colon lysates of C57BL/6 and *p47phox−/−* mice subjected to DSS-induced colitis for a week and treated i.p. with 5 mg/kg Tβ4 for 7 consecutive days after DSS treatment. Normalization was performed on mouse β-actin or Gapdh and corresponding pixel density is depicted. LC3-II band density was normalized to LC3-I to

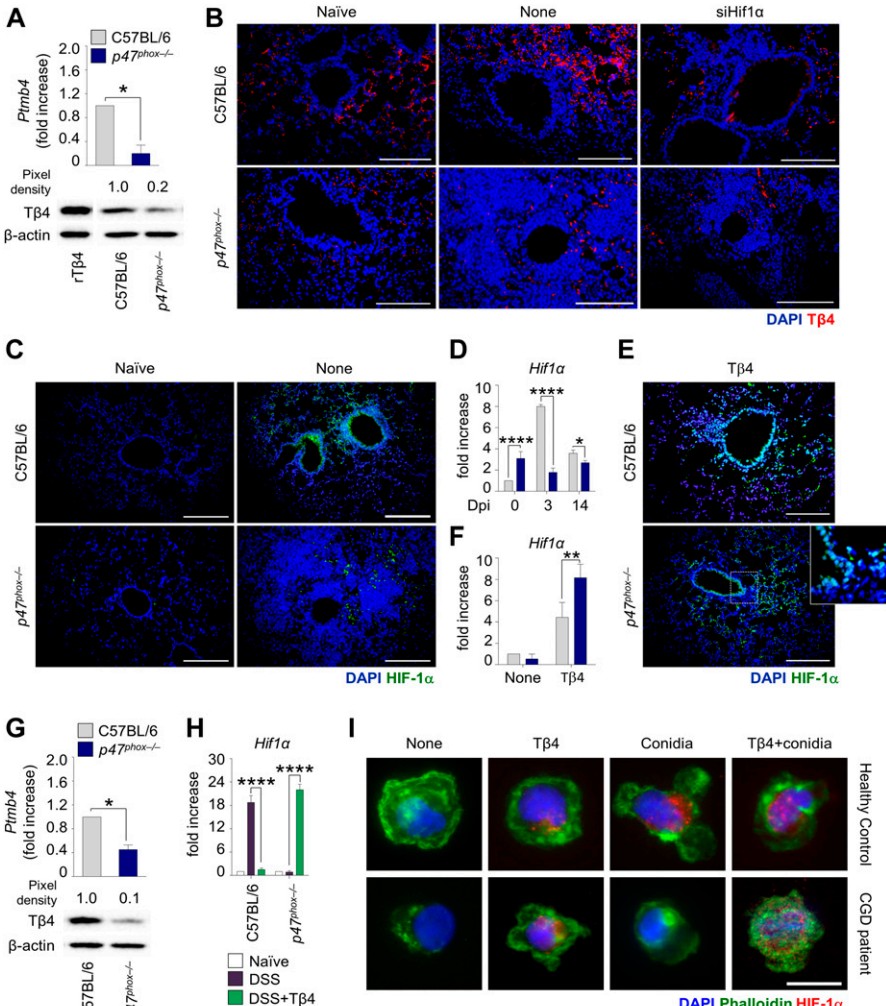

**Figure 2.  Tβ4 promotes HIF-1α expression in CGD.**
**(A, B)** Tβ4 gene expression (*Ptmb4*) and protein levels in the lung of uninfected mice and (B) Tβ4 expression in C57BL/6 and *p47^phox−/−* mice infected intranasally with the fungus and treated with siHif1α. **(C, D)** HIF-1α expression in the lung of infected mice. **(E, F)** HIF-1α expression in the lung of C57BL/6 and *p47^phox−/−* mice infected and treated with Tβ4. Mice were euthanized 7 d after infection. **(G, H)** Tβ4 gene expression (*Ptmb4*) and protein levels, and (H) HIF-1α gene expression in the colon of mice subjected to DSS-induced colitis for a week and treated i.p. with 5 mg/kg Tβ4 for 7 consecutive days after DSS treatment. **(I)** HIF-1α expression on monocytes from CGD patients or healthy controls pretreated with Tβ4 and stimulated for 2 h with the fungus. Gene expression was performed by real-time (RT)-PCR. For immunoblotting, normalization was performed on mouse β-actin, and corresponding pixel density is depicted. Recombinant (r)Tβ4 was used as a positive control. For immunofluorescence, nuclei were counterstained with DAPI. HIF-1α mean fluorescence intensity was measured with ImageJ software. Photographs were taken with a high-resolution microscope (Olympus BX51), 40×, and 100× magnification. For immunofluorescence, data are representative of two independent experiments. For RT PCR, data are presented as mean ± SD of at least two independent experiments. Each independent in vivo experiment includes 6–8 mice per group. *$P < 0.05$, ****$P < 0.0001$, *p47^phox−/−* versus C57BL/6, Tβ4-treated versus untreated (DSS) mice. Unpaired *t* test or two-way ANOVA, Bonferroni post hoc test. Dpi, days post infection; Naïve, uninfected mice. None, control siRNA-treated mice or untreated cells.

## Tβ4 promotes LAP and mucosal barrier protection in an HIF-1α–dependent manner

To assess whether a causal link exists between HIF-1α stabilization and induction of autophagy by Tβ4, we first observed that Tβ4 induced the expression in vitro of *Bnip3* and *Bnip3l,* known to be involved in hypoxia-induced autophagy (Bellot et al, 2009) (Fig S2A). To assess this in vivo, we infected *p47^phox−/−* mice with *A. fumigatus* intranasally and treated them with Tβ4 in the presence or absence of siRNA for HIF-1α. The restoration of LC3-II (Fig S2B) expression in *p47^phox−/−* mice by Tβ4 was abrogated by HIF-1α inhibition, thus indicating that Tβ4 requires HIF-1α to induce LAP. Moreover, as HIF-1α is also directly involved in mucosal barrier protection in hypoxia (Campbell & Colgan, 2019), this would predict a role for Tβ4 in mucosal protection. To prove this, we selected genes known to be regulated by HIF-1α and measured their levels in vivo after treatment with Tβ4. Our screening revealed that

genes involved in angiogenic signaling (*Angpt2, Tie2,* and *Vegfa*), remodeling (*Fgf2*), hormonal regulation (*Epo*), and cell migration (*Cxcr4*) were all up-regulated in the lungs of *Aspergillus*-infected mice upon treatment with Tβ4 (Fig S2C). A similar up-regulation occurred in colons of *p47^phox−/−* mice in the DSS-induced colitis model (Fig S2D). Strikingly, treatment with HIF-1α siRNA completely abrogated the up-regulation induced by Tβ4 in the lung (Fig S2C). Overall, our results indicate that HIF-1α mediates fundamental effects of Tβ4, in the lung and likely in the gut, including LAP and induction of genes involved in the angiogenesis and repair, thus pointing to the Tβ4-HIF-1α axis as a potential therapeutic pathway in CGD.

## Tβ4 ameliorates tissue and immune pathologies in CGD mice

These results would anticipate a beneficial effect of Tβ4 in diseased CGD mice. To this purpose, we evaluated the effect of Tβ4 in mice

---

obtain the ratio. For immunofluorescence, nuclei were counterstained with DAPI. Photographs were taken with a high-resolution microscope (Olympus BX51), 40×, and 100× magnification. LC3 mean fluorescence intensity was measured with ImageJ software. Data are representative of two independent experiments. Each independent in vivo experiment includes 6–8 mice per group. Data are presented as mean ± SD. Naïve, uninfected, or untreated mice; None, infected mice.

with aspergillosis and colitis. The $p47^{phox-/-}$ mice are known to be highly susceptible to pulmonary aspergillosis and inflammation (Romani et al, 2008). Mice were monitored for fungal growth, antifungal activity of effector cells, survival, lung histopathology, and innate and adaptive Th immunity. Thymosin β4 reduced the fungal growth in the lung of both types of mice (Fig 3A), an effect to which the ability of Tβ4 to potentiate phagocytosis and fungal killing of effector phagocytes likely contributed (Fig 3B), and significantly increased the survival of infected mice, being more that 50% of mice survived at the time of which all the untreated mice have died (Fig 3C). Impressively, in $p47^{phox-/-}$ mice, gross lung pathology and histological examination (Fig 3D) revealed no signs of inflammatory lung injury and granuloma formation after Tβ4 administration. Conversely, Tβ4 deficiency by means of siTβ4 administration promoted lung pathology in C57BL/6 mice (Fig S1B). Consistent with the finding that resistance to infection could be restored in these mice by dampening inflammation through NLRP3/IL-1β blocking (de Luca et al, 2014; Iannitti et al, 2016), Tβ4 down-regulated NLRP3 expression in these mice (Fig 3E) and, accordingly, reduced IL-1β, along with TNF-α, IL-17A, and increased IFN-γ production, an effect negated upon siHif1α treatment (Fig 3F). Pathogenic Th2/Th17/Th9 cell responses were down-regulated and protective Th1/Treg cell responses promoted upon Tβ4 treatment (data not shown). Strikingly, the effects on tissue pathology (Fig 3D) and inflammasome expression (Fig 3E) were all abolished by treatment with HIF-1α siRNA, further strengthening the relevance of HIF-1α in mediating Tβ4 effects. These results suggest that Tβ4 ameliorates inflammation and granuloma formation in the CGD lung via HIF-1α. In the murine colitis model, mice were evaluated a day after Tβ4 treatment for weight loss, colon histology, cytokine levels, and tight junction gene expression. Consistent with previous findings (de Luca et al, 2014; Falcone et al, 2016), $p47^{phox-/-}$ mice lost more weight than C57BL/6 mice (about 50% loss of their initial body weight on day 14) (Fig 4A) and had more severe colitis as observed by significantly increased disease activity index scores (Fig 4B). Hematoxylin and eosin staining of colon sections showed severe patchy inflammation characterized by transmural lymphocytic infiltrates, epithelial ulceration, and complete crypt loss (Fig 4C). In addition, $p47^{phox-/-}$ mice displayed high levels of NLRP3 expression (Fig 4D) and IL-1β production (Fig 4E), along with high levels of myeloperoxidase (MPO), TNF-α, and IL-17A (Fig 4F) and low levels of TGF-β (Fig 4G). Treatment with Tβ4 significantly led to weight regain (Fig 4A), decreased disease activity index scores (Fig 4B), amelioration of inflammatory pathology and tissue architecture (Fig 4C), decreased NLRP3 expression (Fig 4D) and inflammatory cytokine levels (Fig 4E and F), and up-regulation of the anti-inflammatory cytokines (Fig 4G). Of interest, Tβ4 greatly promoted the expression of both *Cldn1* and *Ocln*, tight junction proteins that regulate intestinal permeability (Gunzel & Yu, 2013; Kyoko et al, 2014) (Fig 4H), thus suggesting a positive effect on the mucosal barrier function. The protective effect of Tβ4 also occurred when treatment was given concomitantly with DSS (that is, from day 0 to day 7 [Fig S3]), a finding suggesting that prophylactic Tβ4 is also beneficial. Altogether, these results suggest that Tβ4, by activating LAP-DAPK1 and inhibiting inflammasome activity, could have beneficial effects on the outcome of colitis in CGD.

## HIF-1α stabilization recapitulates the effects of Tβ4

Because HIF-1α mediates the effects of Tβ4 in CGD, we sought to investigate whether the stabilization of HIF-1α independent of Tβ4 could similarly exert beneficial effects. For this reason, given the well-known beneficial effects of HIF-1α stabilization on disease outcomes and barrier function in animal models of intestinal inflammation (Colgan et al, 2015), we treated $p47^{phox-/-}$ mice with aspergillosis with dimethyloxalylglycine (DMOG)—a cell-permeable competitive inhibitor of prolyl hydroxylase (PHD) that stabilizes HIF-1α (Mole et al, 2003)—for 5 d. Similar to Tβ4, DMOG reduced fungal burden (Fig 5A), ameliorated lung pathology (Fig 5B), increased HIF-1α expression (Fig 5C), and up-regulated HIF-1α-responsive genes (Fig 5D). Thus, HIF-1α stabilization could be a therapeutic target in CGD.

# Discussion

The results of the present study show that Tβ4 restored autophagy and up-regulated hypoxia-responsive genes in human and murine CGD and this resulted in amelioration of disease pathology (Fig 5E). The increased autophagy, epithelial barrier protection, and repair induced by Tβ4 are consistent with its antioxidative and anti-apoptotic effects (Kumar & Gupta, 2011) and represents a plausible mechanism through which inflammation and granuloma formation are balanced in CGD by Tβ4. It is clear that the innate immune system is pivotal in orchestrating granuloma formation in response to microbial and foreign body challenge (Petersen & Smith, 2013). Impaired antibacterial autophagy links inflammation to granuloma formation in intestinal diseases (Lassen & Xavier, 2018). By promoting LAP, Tβ4 may successfully contribute to pathogen elimination, thus clearly preventing granuloma formation. Thus, the beneficial activity of Tβ4 may encompass a possible activity on the microbial composition at different body sites. Moreover, by inducing TGF-β (Sosne et al, 2004)—known to modulate the fibrotic repair process accompanying granuloma healing (Limper et al, 1994)—Tβ4 may impair granuloma formation. However, excess TGF-β activity, by preventing effective granuloma formation, may interfere with antimicrobial mechanisms (Toossi et al, 1995). Thus, whether the high levels of TGF-β (Toossi et al, 1995) and Tβ4 (Kang et al, 2014) observed in granulomatous lung lesions or colorectal cancer (Gemoll et al, 2015) are the cause or effect of defective granuloma formation needs clarification.

In lung aspergillosis, and likely in DSS-induced colitis, the effects of Tβ4 were dependent on HIF-1α that mediated not only the induction of autophagy but also the up-regulation of hypoxia-responsive genes. Interestingly, Tβ4 up-regulated HIF-1α–dependent genes, involved in barrier protection and angiogenesis and not in glycolysis, are known to contribute to inflammation in myeloid cells (Palazon et al, 2014) (Fig S4A), thus suggesting the unique ability of Tβ4 to activate physiologic HIF-1α to resolve inflammation. HIF-1α can be regulated by both oxygen-dependent and oxygen-independent mechanisms in hypoxic and normoxic conditions, respectively, and strategies are being studied to either activate or inhibit the activity of HIF-1α depending on the clinical setting (Giaccia et al, 2003; Yee Koh et al, 2008). Indeed, whereas HIF-1α inhibition is recognized as an antitumor

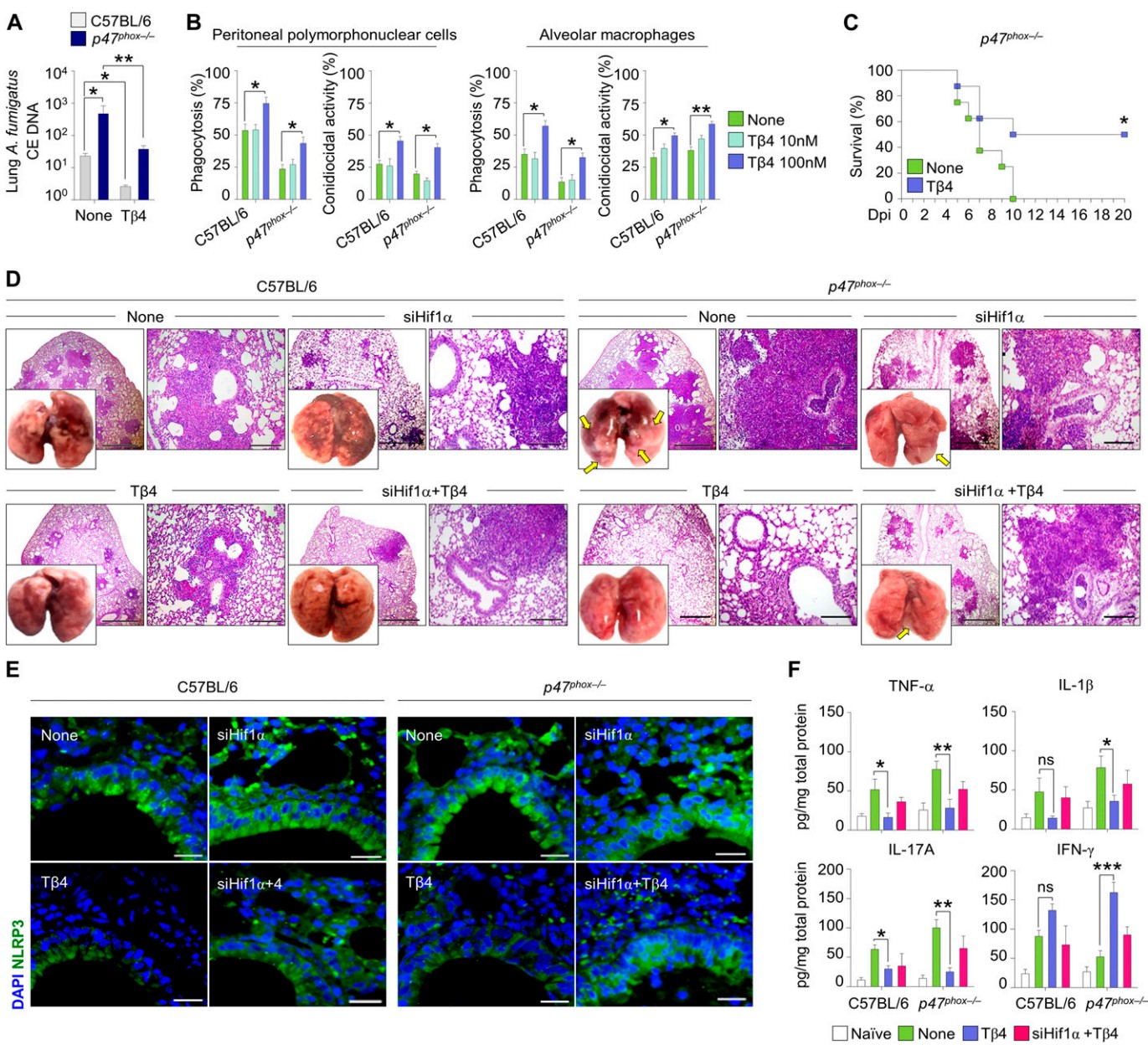

**Figure 3. Tβ4 ameliorates tissue and immune pathologies in CGD mice.**
**(A)** Lung fungal growth of C57BL/6 and $p47^{phox-/-}$ mice infected intranasally with *A. fumigatus* conidia and treated i.p. with 5 mg/kg Tβ4 for 7 consecutive days starting a week after the infection. **(B)** Percent of phagocytosis and conidiocidal activity on peritoneal PMN cells and alveolar macrophages from uninfected C57BL/6 and $p47^{phox-/-}$ mice pre-exposed to different doses of Tβ4 for 1 h before 2 h of pulsing with live *Aspergillus* conidia. **(C)** Survival curves for $p47^{phox-/-}$ mice infected intranasally with *A. fumigatus* conidia and treated for 5 d with Tβ4. **(D, E, F)** Lung gross pathology and histology (periodic acid-Schiff [PAS] staining), (E) NLRP3 expression and (F) cytokines production on lung homogenates of infected mice treated with Tβ4 or siHif1α. For immunofluorescence, nuclei were counterstained with DAPI. Photographs were taken with a high-resolution microscope (Olympus BX51), 4×, 20×, and 40× magnification. Secreted cytokines were assayed by ELISA from supernatants. For histology and immunofluorescence, data are representative of two independent experiments. For RT PCR and ELISA, data are presented as mean ± SD of at least two independent experiments. Each independent in vivo experiment includes 6–8 mice per group. *$P < 0.05$, **$P < 0.01$, ***$P < 0.001$, $p47^{phox-/-}$ versus C57BL/6 mice, Tβ4-treated versus untreated (None) mice or cells. Statistical analyses of the survival curves were performed using the log-rank (Mantel–Cox) test. Two-way ANOVA, Bonferroni post hoc test. Dpi, days post infection; None, control siRNA-treated mice; Naïve, uninfected mice; ns, not significant.

therapeutic strategy, up-regulation of HIF-1α is critical in the treatment of ischemic states (Giaccia et al, 2003; Yee Koh et al, 2008). In the context of CGD, HIF-1α was underexpressed in the gastrointestinal mucosa and lung, suggesting that a potential therapy for CGD should include the elevation of HIF-1α levels to restore the

hypoxia-mediated tissue homeostasis and the optimal antimicrobial response. It has indeed been reported that hypoxia and HIF proteins are required for protection against *Pseudomonas aeruginosa* in vitro (Schaible et al, 2013) and against *A. fumigatus* in vivo (Shepardson et al, 2014).

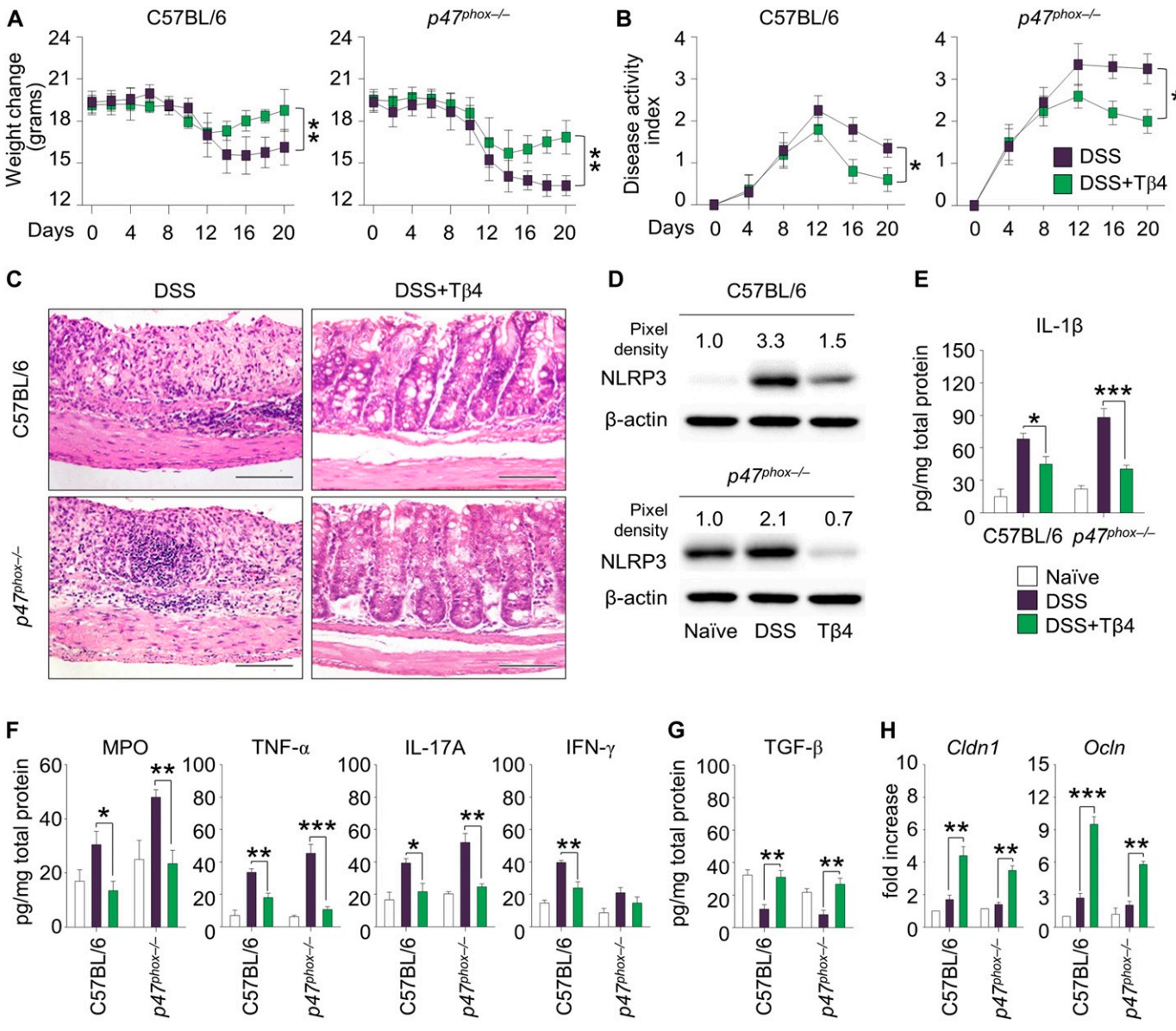

**Figure 4. Tβ4 protects mice with CGD from DSS-induced colitis.**
C57BL/6 and *p47phox−/−* mice were subjected to DSS-induced colitis for a week and treated i.p. with 5 mg/kg Tβ4 for 7 consecutive days after DSS treatment. **(A, B, C, D, E, G, H)** A day after Tβ4 treatment, mice were evaluated for (A) weight change, (B) clinical disease activity index, (C) histological assessment of colitis severity (hematoxylin and eosin staining), (D) NLRP3 protein expression in colon, (E, F) levels of proinflammatory cytokines in colon homogenates, (G) TGF-β production, and (H) *Cldn1* and *Ocln* expression in the colon. Secreted cytokines were assayed by using ELISA from supernatants. Gene expression was performed by RT PCR. For immunoblotting, normalization was performed on mouse β-actin, and corresponding pixel density is depicted. Images were taken with a high-resolution microscope (Olympus BX51), 40× magnification. For histology and immunoblotting, data are representative of two independent experiments. For RT PCR and ELISA, data are presented as mean ± SD of at least two independent experiments. Each independent in vivo experiment includes 10 mice per group. *$P < 0.05$, **$P < 0.01$, ***$P < 0.001$, Tβ4-treated versus untreated (DSS) mice. Two-way ANOVA, Bonferroni or Tukey post hoc test. Naïve, untreated mice.

The dependency of Tβ4 activity on HIF-1α in lung aspergillosis, and likely in DSS-induced colitis, underpins a more general relationship between the two molecules. In agreement with the published literature (Oh et al, 2008; Ryu et al, 2014), we found that Tβ4 and HIF-1α crossregulate their reciprocal expression being the levels of HIF-1α defective in CGD mice but restored by Tβ4 and, conversely, silencing of HIF-1α in wild-type mice being associated with reduced levels of Tβ4. Several mechanisms might underlie the ability of Tβ4 to regulate HIF-1α expression. Thymosin β4 may inhibit PHDs that target

HIF-1α to degradation. Prolyl hydroxylases can be inhibited through succinate upon accumulation by inhibition of succinate dehydrogenase (Selak et al, 2005). Interestingly, in the mouse model of lung inflammation, Tβ4 increased the levels of *Irg1* (Fig S4B), the enzyme responsible for the production of the anti-inflammatory mediator itaconate, an inhibitor of succinate dehydrogenase (Lampropoulou et al, 2016), thus raising the intriguing possibility that Tβ4, by regulating succinate levels, might modulate the levels of HIF-1α via PHDs. However, it has been reported that itaconate may also down-regulate

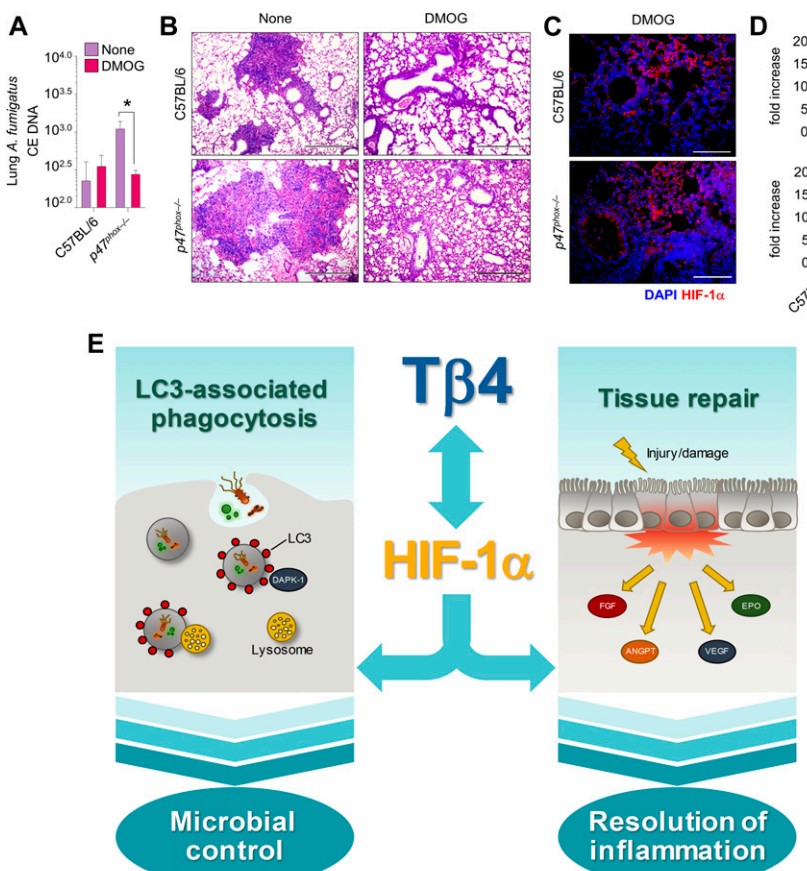

**Figure 5. HIF-1α stabilization recapitulates the effects of Tβ4 in aspergillosis.**
C57BL/6 and *p47^phox−/−* mice were infected intranasally with *A. fumigatus* conidia and treated i.p. with 8 mg/mouse DMOG for 5 d. **(A, B, C, D)** Mice were evaluated for (A) fungal growth, (B) lung histology (PAS staining), (C) HIF-1α protein expression, and (D) *Vegfa* and *Fgf2* expression in the lung. Photographs were taken with a high-resolution microscope (Olympus BX51), 10× and 40× magnification. Gene expression was performed by RT PCR. For immunofluorescence, nuclei were counterstained with DAPI. The control group (none) for HIF-1α immunofluorescense staining is provided in Fig 2C. For histology and immunofluorescence, data are representative of two independent experiments. For RT PCR, data are presented as mean ± SD of at least two independent experiments. Each independent in vivo experiment includes six mice per group. *$P < 0.05$, ***$P < 0.001$, DMOG-treated versus untreated (none) mice. Two-way ANOVA, Bonferroni post hoc test. **(E)** The roposed model of the reciprocal regulation between Tβ4 and HIF-1α in promoting microbial control through LC3-associated phagocytosis and resolution of inflammation in CGD.

HIF-1α levels (Lampropoulou et al, 2016). Alternatively, Tβ4 might stabilize HIF-1α through the production of mitochondrial reactive oxygen species (mtROS), as previously reported (Oh & Moon, 2010). Indeed, we could detect production of mtROS from isolated alveolar macrophages challenged with conidia and treated with Tβ4 (Fig S4C and D), thus supporting the hypothesis that mtROS might mediate the regulation of HIF-1α levels by Tβ4. The ability of Tβ4 to induce mtROS is reminiscent of the activity of pioglitazone, a peroxisome proliferator–activated receptor γ (PPARγ) agonist, currently approved for the treatment of diabetes mellitus type 2 and recently proposed as a therapeutic molecule for CGD (Migliavacca et al, 2016). In normal phagocytes, PPARγ activation linked NADPH oxidase activity with enhanced mtROS production (Fernandez-Boyanapalli et al, 2010), but this mechanism was defective in CGD. Peroxisome proliferator–activated receptor γ agonists, such as pioglitazone, could bypass the need for the NADPH oxidase for enhanced mtROS production and partially restored host defense in CGD. Given that pioglitazone is currently in a phase1/phase2 trial in CGD patients with severe infection (https://clinicaltrials.gov/ct2/show/NCT03080480), it will be intriguing to assess whether endogenous Tβ4 is involved in the clinical efficacy of pioglitazone.

Whatever the mechanism by which Tβ4 regulates HIF-1α expression, the recent finding that HIF-1α reprograms ILC3 metabolism for mucosal barrier protection (Di Luccia et al, 2019), offers a further plausible explanation for the protective function of Tβ4 at the mucosal level.

In conclusion, by unraveling the biological activity of Tβ4 in CGD, this study points to the relevance of mtROS production and HIF-1α stabilization as druggable pathways promoting autophagy and repair in CGD.

# Materials and Methods

## Ethics statement

Patients and healthy volunteers gave written consent to participate as approved by the pediatric hospital Bambino Gesù Institutional Review Board. Murine experiments were performed according to the Italian Approved Animal Welfare Authorization 360/2015-PR and Legislative degree 26/2014 regarding the animal license obtained by the Italian Ministry of Health lasting for 5 yr (2015–2020).

## Cell culture and treatments

RAW264.7 cells (American Type Culture Collection) were grown in supplemented Roswell Park Memorial Institute (RPMI) medium as described (De Luca et al, 2012). Cells were exposed to 10 or 100 nM of Tβ4 (RegeneRx Biopharmaceuticals) for 2 and 4 h at 37°C in 5% $CO_2$ or pretreated for 1 h with Tβ4 at the same concentration before 2 h pulsing with live *A. fumigatus* conidia or inert beads (LB30; Sigma-Aldrich).

Alveolar macrophages from the lung of C57BL/6 and $p47^{phox-/-}$ uninfected mice were obtained after 2 h of plastic adherence at 37°C. Cells were treated as above and evaluated for cellular autophagy markers. Monocytes were isolated from PBMC of healthy donors or two CGD patients, harbouring the mutations c.736C>T, p.Q246X, and whole CYBB gene deletion (69, 84 kb), after informed consent, as described (De Luca et al, 2012). Cells were assessed for LC3 and HIF-1α expression by immunofluorescence.

## Mice

The 6- to 8-wk C57BL/6 (wild type, WT) mice were purchased from Charles River (Calco). Genetically engineered homozygous $p47^{phox-/-}$ mice were bred at the Animal Facility of Perugia University, Perugia, Italy.

## Fungal infection and treatments

Viable conidia from the *A. fumigatus* Af293 strain were obtained as described (De Luca et al, 2012). Mice were anesthetized in a plastic cage by inhalation of 3% isoflurane (Forane Abbott) in oxygen before intranasal instillation of $2 \times 10^7$ resting conidia/20 µl saline. For survival curves, $p47^{phox-/-}$ mice were challenged with $3 \times 10^9$ conidia/20 µl saline. Thymosin β4 was administered i.p. at a dose of 5 mg/kg as an effective dose as described (Badamchian et al, 2003), every day in concomitance with (days 0→7) or after (days 7→14) infection. Dimethyloxalylglycine (Merck Millipore) was administered i.p. at a dose of 8 mg/mouse concomitantly to the infection. For *Hif1a* (duplex name mm.Ri.-Hif1a.13.1; 5′-GAUAUGUUUACUAAAGGACAAGUCA-3′; 3′-UACUAUA-CAAAUGAUUUCCUGUUCAGU-5′) and *Tmsb4x* silencing (duplex name mm.Ri.Tmsb4x.13.1; 5′-CACAUCAAAGAAUCAGAACUACUGA-3′; 3′-AAGUGUAGUUUCUUAGUCUUGAUGACU-5′), each mouse received intranasal administration of 10 mg/kg unmodified siRNA or equivalent dose of nonspecific control siRNA duplex in a volume of 20 µl of duplex buffer (IDT). Intranasal siRNA was given once the day before infection and 1, 3, and 5 d after infection (Iannitti et al, 2013b). It is known that lung-specific siRNA delivery can be achieved by intranasal administration without the use of viral vectors or transfection agents in vivo (Iannitti et al, 2013a). Mice were euthanized 7 or 14 d post-infection. Fungal burden was determined by quantitative PCR and expressed as conidial equivalents. Lung tissue was aseptically removed and homogenized in 3 ml of sterile saline. Lung homogenates were subjected to a secondary homogenization step with 0.5 mm glass beads in Bead Beater homogenizer (Gemini BV) and then processed for DNA extraction with the QIAamp DNA Mini Kit (QIAGEN) according to the manufacturer's directions. Fungal burden was quantified by quantitative PCR by using the sequences for the multicopy 18S ribosomal DNA gene. For lung histology, sections (3–4 µm) of paraffin-embedded tissues were stained with periodic acid-Schiff.

## DSS-induced colitis

DSS (2.5% wt/vol, 36,000–50,000 kD; MP Biomedicals) was administered in drinking water ad libitum for 7 d. Fresh solution was replaced on day 3. Mice were injected with 5 mg/kg of Tβ4 every day i.p. in concomitance with (days 0→7) or after (days 7→14) DSS administration. The control received the diluent alone. Weight loss,

stool consistency, and faecal blood were recorded daily. Upon necropsy (7 and 14 d after DSS administration), tissues were collected for histology and cytokine analysis. Colonic sections were stained with hematoxylin and eosin. The colitis disease activity index was calculated daily for each mouse based on weight loss, occult blood, and stool consistency. A score of 1–4 was given for each parameter as described (McNamee et al, 2011).

## Immunoblotting

For immunoblotting, organs or cells were lysed in Radio-Immuno-precipitation Assay buffer. The lysate was separated in SDS–PAGE and transferred to a nitrocellulose membrane. The membranes were incubated with the following primary antibodies at 4°C overnight: anti-DAPK1 (antibodies-online.com), anti-Rubicon and anti-NLRP3 (Cell Signaling), anti-Tβ4 (Abcam), and anti-LC3B (Novus; Cell Signaling or Abcam). Normalization was performed by probing the membranes with mouse anti-β-actin and anti-Gapdh antibodies (Sigma-Aldrich). Normalization was performed on mouse β-actin or Gapdh, and corresponding pixel density was depicted. LC3-II band density was normalized to LC3-I to obtain the ratio. The ratio of the untreated control was set as one. Chemiluminescence detection was performed with LiteAblot Plus chemiluminescence substrate (EuroClone S.p.A.), using the ChemiDocTM XRS+ Imaging System (Bio-Rad) and quantification was obtained by densitometry image analysis using Image Lab 5.1 software (Bio-Rad).

## Immunofluorescence staining

For immunofluorescence, monocytes from CGD patients or controls were grown in supplemented RPMI and placed on microscope glass slides at 37°C for adhesion. Slides were then washed with PBS and fixed with 4% of paraformaldehyde. Cells were incubated in blocking solution (PBS-3% BSA-0.1% Triton X-100) with anti-LC3B antibody (Nanotools) and anti-HIF-1α (Abcam). After overnight staining with primary antibodies, slides were washed and incubated with anti-IgG and rabbit-TRITC (Sigma-Aldrich). Alexa Fluor 488 phalloidin was used for selective labelling of F-actin. LC3B (Abcam), Tβ4 (ABclonal), HIF-1α, and NLRP3 (Abcam) staining of lung sections were performed as described. Nuclei were counterstained with DAPI. Images were acquired using a fluorescence microscope (BX51; Olympus) and analySIS image processing software (Olympus).

## RT PCR

Real-time PCR was performed using CFX96 Touch RT PCR Detection System and SYBR Green chemistry (Bio-Rad). Organs or cells from pooled mice (n = 6–8 mice/group for lungs and n = 10 mice/group for colons) were lysed, and total RNA was reverse transcribed with PrimeScript RT Reagent Kit with gDNA Eraser (Takara), according to the manufacturer's instructions. The PCR primers sequences (5′-3′) are as follows:

*Ptmb4*: ACAAACCCGATATGGCTGAG and GCCAGCTTGCTTCTCTTGTT
*Hif1a*: TCAAGTCAGCAACGTGGAAG and TTCACAAATCAGCACCAAGC
*Hif1b*: CAAGCATCTTTCCTCACTGATC and ACACCACCCGTCCAGTCTCA
*Cldn1*: AGCCAGGAGCCTCGCCCCGCAGCTGCA and CGGGGTTGCCTGCAAAGT

*Ocln*: GTTGATCCCCAGGAGGCTAT and CCATCTTTCTTCGGGTTTTC
*Vegfa:* CAGGCTGCTGTAACGATGAA and GCATTCACATCTGCTGTGCT
*Fgf2*: CGACCCACACGTCAAACTAC and GCCGTCCATCTTCCTTCATA
*Bnip3*: GCTCCCAGACACCACAAGAT and TGAGAGTAGCTGTGCGCTTC
*Bnip3l*: CCTCGTCTTCCATCCACAAT and GTCCCTGCTGGTATGCATCT
*Angpt2*: GAACCAGACAGCAGCACAAA and TGGTCTGATCCAAAATCTGCT
*Tie2*: CGGCCAGGTACATAGGAGGAA and TCACATCTCCGAACAATCAGC
*Epo*: ACTCTCCTTGCTACTGATTCCT and ATCGTGACATTTTCTGCCTCC
*Cxcr4*: GGGTCATCAAGCAAGGATGT and GGCAGAGCTTTTGAACTTGG
*Dapk1*: CCTGGGTCTTGAGGCAGATA and TCGCTAATGTTTCTTGCTTGG
*Ldha*: AGGCTCCCCAGAACAAGATT and TCTCGCCCTTGAGTTTGTCT
*Pkm*: CGATCTGTGGAGATGCTGAA and AATGGGATCAGATGCAAAGC
*Glut1*: GCTGTGCTTATGGGCTTCTC and CACATACATGGGCACAAAGC
*Irg1*: AGTTCCAACACCTCCAGCAC and GGTGCCATGTGTCATCAAAA

Amplification efficiencies were validated and normalized against *Gapdh*. The thermal profile for SYBR Green RT PCR was at 95°C for 3 min, followed by 40 cycles of denaturation for 30 s at 95°C, and an annealing/extension step of 30 s at 60°C. Each data point was examined for integrity by analysis of the amplification plot.

### ELISA

To evaluate cytokine production in DSS-induced colitis, colons were opened longitudinally and washed in complete medium with antibiotics and were cultured at 37°C for 24 h in RPMI and 5% FBS. The supernatants were collected for ELISA. The levels of cytokines were determined by using specific ELISAs (R&D Systems) in accordance with the manufacturer's protocols. The concentration of secreted cytokines in the colon supernatants or lung homogenates was normalized to total tissue protein by using Quant-iT Protein Assay Kit (Life Technologies). Results are expressed as picogram of cytokine per microgram of total protein. The myeloperoxidase (MPO) content in colonic tissues were determined using commercially available kits (Nanjing Jiancheng Bioengineering Institute).

### Antifungal effector activity

Murine polymorphonuclear (PMN) cells from C57BL/6 or *p47phox−/−* uninfected mice were positively selected with magnetic beads (Miltenyi Biotec) from the peritoneal cavity of mice 8 h after the intraperitoneal injection of 1 ml endotoxin-free 10% thioglycolate solution. On FACS analysis, Gr-1+PMNs were 98% pure and stained positive for the CD11b myeloid marker. Monolayers of plastic-adherent macrophages were obtained, after 2 h plastic adherence, from the lung of C57BL/6 and *p47phox−/−* uninfected mice. Cells were pretreated for 1 h with different concentrations of Tβ4 (10 and 100 nM) before pulsing with *A. fumigatus* conidia (1:3 cell:fungus for phagocytosis and 10:1 cell:fungus for conidiocidal activity) for 120 min at 37°C. The percentage of CFU inhibition (mean ± SD) was determined as described previously (Bellocchio et al, 2004).

### ROS determination

Alveolar macrophages from the lung of C57BL/6 and *p47phox−/−* uninfected mice were assessed for intracellular ROS production by dihydrorhodamine 123 (DHR) evaluation. As an inhibitor, we used

MitoTEMPO to scavenge mitochondrial ROS. For ROS determination, 10 $\mu$M DHR (Sigma-Aldrich) was added to cells exposed to 100 nM Tβ4, 10 ng/ml PMA (phorbol 12-myristate 13-acetate) (Sigma-Aldrich), and/or *A. fumigatus* conidia at cell:fungus of 1:1 for 1 h at 37°C. Cells were plated on a 96-well culture plate in HBSS buffer with Ca$^{2+}$ and Mg$^{2+}$ but without phenol red. Cells were preincubated with 50 $\mu$M MitoTEMPO (Enzo Life Science) for 1 h before the addition of Tβ4. The DHR was measured by the multifunctional microplate reader Tecan Infinite 200 (Tecan) at different time points. The results expressed as relative fluorescence units are the means ± SD of at least two experiments in duplicate.

### Statistical analysis

GraphPad Prism 6.01 program (GraphPad Software) was used for analysis. Data are expressed as mean ± SD. Statistical significance was calculated by using two-way ANOVA (Tukey or Bonferroni post hoc test) for multiple comparisons and unpaired *t* test for single comparisons. Statistical analysis of the survival curves was performed using the log-rank (Mantel–Cox) test. The variance was similar in the groups being compared. Cell fluorescence intensity was measured by using ImageJ software.

# Supplementary Information

# Acknowledgements

This study was supported by the Specific Targeted Research Project FunMeta (ERC-2011-AdG-293714 to L Romani). MM Bellet was supported by a fellowship from Fondazione Umberto Veronesi.

## Author Contributions

G Renga: conceptualization, data curation, supervision, investigation, and methodology.
V Oikonomou: conceptualization, data curation, supervision, investigation, and methodology.
S Moretti: conceptualization and data curation.
C Stincardini: investigation and methodology.
MM Bellet: supervision and writing—original draft.
M Pariano: investigation and methodology.
A Bartoli: supervision and writing—original draft.
S Brancorsini: investigation and methodology.
P Mosci: investigation and methodology.
A Finocchi: resources.
P Rossi: resources.
C Costantini: supervision and writing—original draft, review, and editing.
E Garaci: supervision and writing—original draft.
AL Goldstein: supervision and writing—original draft.

L Romani: conceptualization, supervision, funding acquisition, project administration, and writing—original draft, review, and editing.

## Conflict of Interest Statement

One of the authors AL Goldstein is chairman of the board and a holder of stock in RegeneRX Biopharmaceuticals, Inc., a company developing T$\beta$4 for the clinic. The other authors declare no competing financial interests.

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
