## [Reviewer comments · Life Science Alliance]

Life Science Alliance

Thymosin β 4 promotes autophagy and repair via HIF-1 α stabilization in chronic granulomatous disease

Giorgia Renga, Vasilis Oikonomou, Silvia Moretti, Claudia Stincardini, Marina Bellet, Marilena Pariano, Andrea Bartoli, Stefano Brancorsini, Paolo Mosci, Andréa Finocchi, Paolo Rossi, Claudio Costantini, Enrico Garaci, Allan Goldstein, and Luigina Romani

DOI: <https://doi.org/10.26508/lsa.201900432>

Corresponding author(s): Luigina Romani, University of Perugia; Giorgia Renga, University of Perugia; and Vasilis Oikonomou, University of Perugia

Review Timeline:

Submission Date:	2019-05-20
Editorial Decision:	2019-07-10
Revision Received:	2019-10-11
Editorial Decision:	2019-10-28
Revision Received:	2019-10-31
Accepted:	2019-11-04

Scientific Editor: Andrea Leibfried

Transaction Report:

July 10, 2019

Re: Life Science Alliance manuscript #LSA-2019-00432-T

Prof. Luigina Romani
University of Perugia
Perugia
Italy

Dear Dr. Romani,

Thank you for submitting your manuscript entitled "Thymosin β 4 promotes autophagy and repair via HIF-1 α stabilization in chronic granulomatous disease" to Life Science Alliance. The manuscript was assessed by expert reviewers, whose comments are appended to this letter.

As you will see, the reviewers appreciate your data. However, they would expect more definitive insight into the functional significance of your findings and into the potential causality of the described relationships. Should you be able to provide such insight, we would be happy to invite you to submit a revised version of your manuscript to us. Importantly, the requested quantifications, controls and statistical analyses should get provided, the physiological relevance of TB4 and the potential link between TB4 and Hif-1a should get further tested.

Thank you for this interesting contribution to Life Science Alliance. We are looking forward to receiving your revised manuscript.

Sincerely,

B. MANUSCRIPT ORGANIZATION AND FORMATTING:

Reviewer #2 (Comments to the Authors (Required)):

In this manuscript from Renga et al, they examine the role TB4 in enhancing LAP in CGD mice to ameliorate *Aspergillus fumigatus* infections and colitis in those mice. The data present suggest that TB4 treated of p47 KO mice can ameliorate fungal disease and colitis symptoms. The data also suggests that this occurs in a HIF1alpha-dependent manner. This is an interesting observation, but

the manuscript lacks some key controls to tie everything together. Additionally, the question of whether TB4 acts normally in the setting of WT cells/mice remains important unaddressed question. Is this physiologically important for fungal control or occurring just in the setting of pharmacological provision of TB4?

SPECIFIC POINTS:

- 1) Figure 1: The authors needs to explain how the Western Blot "ratio" and "pixel density" was calculated. Especially for the "ratio", is this normalized, because the untreated groups in panels A & B look very different but the ratio in both is 1.0, why is this?
- 2) Figure 2E needs to be quantified like done in Figure 2C.
- 3) Figure 3: Panel A lacks statistical analysis between the WT and p47 KO groups. In panel C, the authors should show the exact same 3 types of pictures for the siHIF1a group (gross lungs, zoomed out H&E, and zoomed in lesional H&E). In panel E, are the cytokine differences loss by siHIF1a treatment of the TB4 group?
- 4) Figure 4: Is the amelioration of colitis symptoms by TB4 treatment also dependent of Hif-1a?
- 5) Figure 5: Panel A lacks statistical analysis so the authors cannot claim changes in fungal burden. Moreover, similar to Figure 3, are there changes in the phagocytotic and killing ability of peritoneal polymorphonuclear cells and/or alveolar macrophages. Panel C lacks IHC staining of untreated controls
- 6) Overall, is the role of TB4 physiologically relevant. Does the loss of TB4 in WT mice result in a reduction of fungal-induced LAP and loss of control of fungal infection?

Reviewer #3 (Comments to the Authors (Required)):

- 1) In this manuscript, the authors describe the results of a study of the effects of thymosin B4 on non-canonical autophagy in macrophages and HIF-1a expression, inflammation, and granuloma formation in CGD mice with colitis or aspergillosis. Their results clearly show that TB4 treatment promoted macrophage autophagy and resolution of inflammation in colitis and aspergillosis. The results show the potential for TB4 therapy in treatment of patients with detrimental inflammation in infection and disease.
- 2) The data are strongly supportive of the conclusions, which the authors are careful not to overstate.
- 3) The Results and Discussion in the manuscript could be clarified in some areas:
 - i) Fig. 1. One can assume the ratio of LC3-II to LC3-I is a proxy for levels of autophagy, however, this should be explicitly stated/explained.
 - ii) Macrophages and monocytes are used exclusively in the manuscript. Why are neutrophils not examined? This may be obvious to the authors, but not to this reviewer, and it may be helpful to explain the rationale for the uninformed.
 - iii) Was there no effect of TB4 on survival? Survival is not examined in any model.
 - iv) in Fig. 3E the results for IFN γ do not match the description of the results. It seems as if that panel should be labeled IL-10. If so, then the panel for IFN γ should be added.

v) It might be helpful to add a model figure to show the interactions of TB4, autophagy, HIF-1a, and inflammatory pathways in CGD. It is difficult for this reviewer to picture these associations, and a model figure may serve to orient the reader to the authors' interpretation of the results.

Reviewer #2 (Comments to the Authors (Required)):

In this manuscript from Renga et al, they examine the role TB4 in enhancing LAP in CGD mice to ameliorate *Aspergillus fumigatus* infections and colitis in those mice. The data present suggest that TB4 treated of p47 KO mice can ameliorate fungal disease and colitis symptoms. The data also suggests that this occurs in a HIF1 α -dependent manner. This is an interesting observation, but the manuscript lacks some key controls to tie everything together. Additionally, the question of whether TB4 acts normally in the setting of WT cells/mice remains important unaddressed question. Is this physiologically important for fungal control or occurring just in the setting of pharmacological provision of TB4?

SPECIFIC POINTS:

1) Figure 1: The authors needs to explain how the Western Blot "ratio" and "pixel density" was calculated. Especially for the "ratio", is this normalized, because the untreated groups in panels A & B look very different but the ratio in both is 1.0, why is this?

Author's response. We apologize for the inaccuracy. In Figure 1B untreated group ("0") refers to inert beads which have been used as a control for LAP evaluation (Oikonomou et al, 2016). We have now better explained in the Material and Methods and Figure 1 legend the ratio and pixel density.

2) Figure 2E needs to be quantified like done in Figure 2C.

Author's response. Figure 2E has now quantification like in Figure 2C.

3) Figure 3: Panel A lacks statistical analysis between the WT and p47 KO groups. In panel C, the authors should show the exact same 3 types of pictures for the siHIF1 α group (gross lungs, zoomed out H&E, and zoomed in lesional H&E). In panel E, are the cytokine differences lost by siHIF1 α treatment of the TB4 group?

Author's response. The statistical analysis between the WT and p47 KO groups has been added in Panel A. We added the exact same 3 types of pictures for the siHif1 α group (Panel D). The cytokine levels in siHif1 α -treated mice are now provided (Panel F).

4) Figure 4: Is the amelioration of colitis symptoms by TB4 treatment also dependent of Hif-1 α ?

Author's response. Yes. Fig. S1 panel D show data clearly indicating that the downstream Hif1 α genes were indeed induced by T β 4 treatment in the DSS colitis. As T β 4 lost its activity in Caco-2 cells treated with siHif1 α (our own unpublished data), it is plausible to imply a role for HIF1 α in the effects of T β 4 in the gut.

5) Figure 5: Panel A lacks statistical analysis so the authors cannot claim changes in fungal burden. Moreover, similar to Figure 3, are there changes in the phagocytotic and killing ability of peritoneal polymorphonuclear cells and/or alveolar macrophages. Panel C lacks IHC staining of untreated controls

Author's response. The statistic is now provided in panel A. Being the fungal burden significantly reduced, it is quite likely that DMOG treatment also potentiated the antifungal activity of effector phagocytes. Unfortunately, these effector activities have not been tested. For the IF staining of the untreated controls, please refer to Fig. 2C (specified in the figure legend).

6) Overall, is the role of TB4 physiologically relevant. Does the loss of TB4 in WT mice result in a reduction of fungal-induced LAP and loss of control of fungal infection?

Author's response. Yes. The new Fig. S1A and B clearly shows that the inhibition of endogenous T β 4 resulted in loss of fungal-induced LAP and loss of control of the infection, as revealed by the increased lung immunopathology in infection.

Reviewer #3 (Comments to the Authors (Required)):

1) In this manuscript, the authors describe the results of a study of the effects of thymosin B4 on non-canonical autophagy in macrophages and HIF-1a expression, inflammation, and granuloma formation in CGD mice with colitis or aspergillosis. Their results clearly show that TB4 treatment promoted macrophage autophagy and resolution of inflammation in colitis and aspergillosis. The results show the potential for TB4 therapy in treatment of patients with detrimental inflammation in infection and disease.

2) The data are strongly supportive of the conclusions, which the authors are careful not to overstate.

3) The Results and Discussion in the manuscript could be clarified in some areas:

i) Fig. 1. One can assume the ratio of LC3-II to LC3-I is a proxy for levels of autophagy, however, this should be explicitly stated/explained.

Author's response. Explained (second line of the Results).

ii) Macrophages and monocytes are used exclusively in the manuscript. Why are neutrophils not examined? This may be obvious to the authors, but not to this reviewer, and it may be helpful to explain the rationale for the uninformed.

Author's response. The reviewer may have missed the data in Fig. 3 panel B referring to PMN.

iii) Was there no effect of TB4 on survival? Survival is not examined in any model.

Author's response. The effect of T β 4 on survival is now provided in Fig. 2 panel C

iv) In Fig. 3E the results for IFN γ do not match the description of the results. It seems as if that panel should be labeled IL-10. If so, then the panel for IFN γ should be added.

Author's response. We apologize for the inaccuracy. We have eliminated the incorrect label.

v) It might be helpful to add a model figure to show the interactions of TB4, autophagy, HIF-1a, and inflammatory pathways in CGD. It is difficult for this reviewer to picture these associations, and a model figure may serve to orient the reader to the authors' interpretation of the results.

Author's response. A model figure is now shown in Fig. 5 panel E.

October 28, 2019

RE: Life Science Alliance Manuscript #LSA-2019-00432-TR

Prof. Luigina Romani
University of Perugia
P.le Gambuli, 06132 Perugia, Italy
Perugia 06132
Italy

Dear Dr. Romani,

Thank you for submitting your revised manuscript entitled "Thymosin β 4 promotes autophagy and repair via HIF-1 α stabilization in chronic granulomatous disease". As you will see, a few issues should still get addressed prior to acceptance of your article here. We would thus like to invite you to submit a final version, addressing the following:

- Please address the remaining concerns of the reviewer by text changes
- Please link your ORCID iD to your profile in our submission system, you should have received an email with instructions on how to do so
- Please add a callout in the manuscript text to Fig S1A
- Please describe the blot in Figure 2A in the legend
- Please re-arrange the panels in figure 1E, 2I, 3E, S2B to have the single channel / the individual experiments that the reader should compare side-by-side. The current display suggests that you are showing magnifications of inserts, which is not the case
- Please add the number of replicates in the description for the RT-PCR analysis performed
- Please add the statistical test used directly in the figure legends where you mention the p-values
- Please make sure that the number of replicates is apparent for each figure (eg, see figure 1 legend where n is unclear for panel A-E and I; it is also unclear whether n=x mice refers to the total number of mice or to the number of mice used for each representative result).

A. FINAL FILES:

B. MANUSCRIPT ORGANIZATION AND FORMATTING:

Sincerely,

Andrea Leibfried, PhD
Executive Editor
Life Science Alliance

Meyerhofstr. 1
69117 Heidelberg, Germany
t +49 6221 8891 502
e a.leibfried@life-science-alliance.org
www.life-science-alliance.org

Reviewer #2 (Comments to the Authors (Required)):

I am still concerned about the authors failing to link together the role of Hif1-alpha in the DSS colitis model. Yes, the authors indeed demonstrate that genes typically downstream of Hif1-alpha are elevated in that model (Supp. Figure 2D & 4D), but they do not show those genes are dependent on Hif1-alpha using the siRNA in their model. The authors rebuttal letter states it is "...plausible to imply a role for HIF1a in the effect of TB4 in the gut...", but in their discussion they at the start of their second paragraph of the discussion they state "...effects of TB4 were dependent on HIF1a...". This type of strong wording on the dependency in the DSS model needs to be softened throughout the manuscript, and be more in line with their rebuttal letter, that it is plausible, unless they conduct the siRNA studies in the DSS model too.

Minor Point: In the methods, explicitly showing the formula for the pixel density and LC3 ratio would be helpful. Shouldn't 1.0 mean the two band are of the same intensity or has everything been normalized back to the untreated control (which is set as 1)?

November 4, 2019

RE: Life Science Alliance Manuscript #LSA-2019-00432-TRR

Prof. Luigina Romani
University of Perugia
P.le Gambuli, 06132 Perugia, Italy
Perugia 06132
Italy

Dear Dr. Romani,

Thank you for submitting your Research Article entitled "Thymosin β 4 promotes autophagy and repair via HIF-1 α stabilization in chronic granulomatous disease". It is a pleasure to let you know that your manuscript is now accepted for publication in Life Science Alliance. Congratulations on this interesting work.

*****IMPORTANT:** If you will be unreachable at any time, please provide us with the email address of an alternate author. Failure to respond to routine queries may lead to unavoidable delays in publication.*******

DISTRIBUTION OF MATERIALS:

Again, congratulations on a very nice paper. I hope you found the review process to be constructive and are pleased with how the manuscript was handled editorially. We look forward to future exciting submissions from your lab.

Sincerely,
